



# On the compatibility of Brewer total column ozone measurements in two adjacent valleys (Arosa and Davos) in the Swiss Alps

René Stübi[1], Herbert Schill[2], Jörg Klausen[1], Laurent Vuilleumier[1], Julian Gröbner[3], Luca Egli[3], and Dominique Ruffieux[1]

[1]Federal Office of Meteorology and Climatology, MeteoSwiss, 1530 Payerne, Switzerland
[2]Federal Office of Meteorology and Climatology, MeteoSwiss, Lichtklimatisches Observatorium, 7050 Arosa, Switzerland
[3]Physikalisch-Meteorologisches Observatorium / World Radiation Center, 7260 Davos Dorf, Switzerland

*Correspondence to:* R. Stübi (rene.stubi@meteoswiss.ch)

**Abstract.** The Arosa site is well known in the ozone community for its continuous total ozone column observations recorded since 1926. Originally based on Dobson sun spectrophotometers, the site has been gradually complemented by three automatic Brewer instruments, in operation since 1998. To secure the long term ozone monitoring in this Alpine region and to benefit from synergies with the World Radiation Center, the feasibility of moving this activity to the nearby site at Davos (aerial

distance of 13 km) has been explored. Concerns about a possible rupture of the 90 years long record has motivated a careful comparison of the two sites since great attention to the data continuity and quality has always been central to the operations of the observatory at Arosa. To this end, one element of the Arosa Brewer triad has been set up at the Davos site since November 2011 to realize a campaign of parallel measurements and to study the deviations between the three Brewer instruments. The analysis of the coincident measurements shows that the differences between Arosa and Davos remain within the range of the

long term stability of the Brewer instruments. A non-significant seasonal cycle is observed, which could possibly be induced by a stray light bias and the altitude difference between the two sites. These differences are shown to be lower than the short term variability of the time series and the overall uncertainty from individual Brewer instruments and therefore are not statistically significant. It is therefore concluded that the world's longest time series of the total ozone column obtained at Arosa site could be safely extended and continued with measurements taken from instruments located at the nearby Davos site without

introducing a bias in this unique record.

## 1 Introduction

From the 1920s onwards, good quality ozone column measurements have been obtained by the sun spectrophotometry technique initially developed by Prof. G. Dobson (*Komyhr,*, 1980; *Basher*, 1982; *Komhyr et al.*, 1989). At that time, he created the first ozone network of the so-called Dobson instruments distributed at four sites in Europe, one of them being Arosa. The

good quality of the Arosa measurements was the result of a very involved scientist (Prof. W. P. Götz) carefully maintaining the observations in this high altitude and low polluted environment. This motivated Prof. G. Dobson to permanently leave an instrument for continuous operation at the LichtKlimatisches Observatorium (LKO) Arosa. The LKO had been founded a few years earlier at the initiative of a medical corporation with the initial purpose of understanding why Alpine atmosphere





proved to be beneficial for patients suffering from tuberculosis disease (*Staehelin et al.*, 2016). The long term continuation of the ozone column monitoring at LKO has continued in the 1960s under the responsibility of Prof. H.-U. Dütsch and later on, from the 1980s, under the responsibility of Prof. J. Staehelin associated to the Federal Office of Meteorology and Climatology (MeteoSwiss) to assure the technical support and the development of the LKO. The Arosa ozone column measurements record

is well known in the ozone community as the longest continuous series worldwide starting in 1926 (*Staehelin et al.*, 1998; *Scarnato et al.*, 2009, 2010).

The ozone column time series was initially composed of measurements from a suite of manual and automatic Dobson sun spectrophotometers with various contiguous or overlapping operation periods as illustrated in the lower part of Figure 1 (blue segments). On a regular basis since 1957, the ozone column series have been complemented by ozone profile measurements

based on the Umkehr technique (*Petropavlovskikh et al.*, 2011). From 1994 onwards, three Dobson instruments have been in operation at Arosa, with two instruments focused on ozone column and one on Umkehr profile measurements. In Figure 1, the annual mean ozone column composite time series is illustrated. The ozone hole problem, first observed in the polar regions (*Solomon*, 1999; *Farman et al.*, 1985), has had a signature at mid-latitudes of ∼5% decrease of the total ozone column clearly evident in Arosa between 1970 and 1990 in Figure 1 (blue broken lines). The consequences of the Montreal protocol and its

successive amendments appear as a leveling of the ozone column over the last past decade (*Pawson et al.*, 2014). This first stage of the ozone layer recovery is clear. However the expected increase does not appear unambiguously and statistically significant in the LKO series (*Yang et al.*, 2008).

Developed in Canada, the new generation Brewer sun spectrophotometers have been introduced in the market in the 1980s (*Kerr et al.*, 1981; *Kerr and McElroy*, 1995). These fully automatic Brewer instruments have been gradually introduced at the

LKO over the time period 1988 to 1998 to constitute a triad as illustrated at the top of Figure 1 (red segments) (*Stübi et al.*, 2017).

The Physikalisches Meteorologisches Observatorium (PMOD) located in the adjacent Davos valley was founded in 1907 and played a similar role as the LKO. Today, PMOD is well known for its activities in the domain of radiation measurements and worldwide calibration services and has acquired the status of the World Radiation Center (WRC) for the World Meteorological

Organization (WMO).

The question of pooling the PMOD/WRC and LKO resources has regularly been raised, for profiting from synergies between the two institutions and improving efficiency. The main objection to a change of site for the ozone column monitoring has always been focused on the importance of the long LKO measurements series, which should neither be discontinued nor disrupted. To assure the quality of these important historical Dobson and Brewer series in Switzerland on the long term, a new

analysis of the situation was conducted at MeteoSwiss that led to a two phase multi-year project.

In the first phase, the Dobson instrument was automated resulting in a significant increase of the number of measurements as well as improved data quality. The description of the automated version of the Dobson instrument will be the subject of a separate publication. In parallel to these technical developments, a campaign of parallel Brewer measurements at the two sites Arosa and Davos was initiated to objectively determine any influences of the environment (e.g. altitude, surrounding topography, etc.) and location on the measured ozone column. The results of this campaign are presented in this study. In the




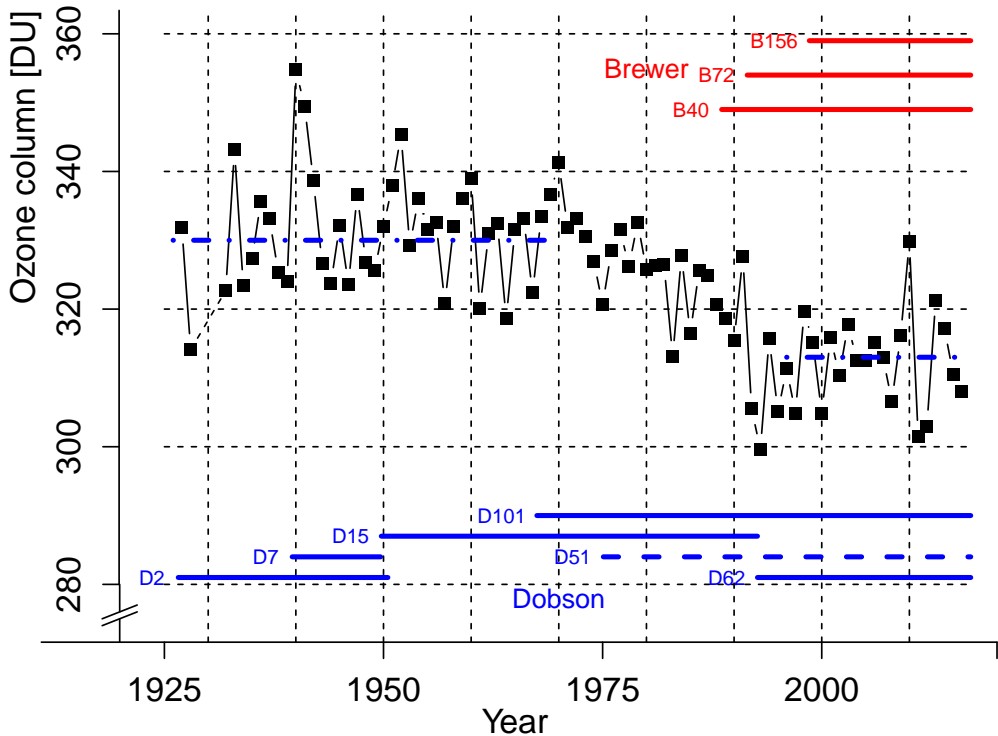

**Figure 1.** Time series of the annual mean ozone column above Arosa, Switzerland. The different instruments in operation are illustrated by blue segments for the Dobson and red segments for the Brewer instruments. Dobson $D_{051}$ is mostly dedicated to Umkehr measurements. The broken blues lines are the mean ozone column before the 1970s, respectively after the 1997s.

second phase, a similar campaign of parallel measurements with automated Dobson instruments is running since January 2016. However, the middle term stability and the calibration status after the major upgrade of these instruments have to be assessed

5 before publishing these results and a 2–3 years comparison campaign is also necessary.

The 13 years of the Arosa Brewer triad covering the previous period 1998–2011 were carefully analyzed to quantify the reproducibility and the long term stability of these Brewer measurements. The results of this latter analysis are described in *Stübi et al.* (2017) and reference to this publication will appear in the present analysis since it is based on a similar approach. The combined information from the two studies would provide the objective scientific evidence on which to base a decision on

10 relocating the Dobson and Brewer instruments from Arosa to Davos.

In section 2, the two measurement sites are briefly presented with emphasis on the environmental factors with a potential to produce systematic effects. In section 3, the instruments and the data sets are described and the analysis of the 2010–2016 measurements period with Brewer $B_{072}$ instrument located first at Arosa (until end of 2011) and then at Davos is presented. The discussion of the results is found in section 4 and followed by the conclusions in section 5.



## 2   Sites Characteristics

Arosa is an alpine resort (∼2500 permanent inhabitants) in the Swiss Alps at a mean altitude of 1800 m a.s.l surrounded by
a mountain circus with summits reaching 3000 m a.s.l. The LKO measurement site (46.779 N, 9.675 E) is on the terrace of
a building at 1850 m a.s.l where the two triads of sun spectrophotometers (Dobson and Brewer) were collocated until the
beginning of the campaign in November 2011. The area is isolated from major industrial pollution sources since Arosa is
connected to the Rhine valley by a ∼30 km long narrow valley and a altitude difference of ∼1000 m preventing the inflow of
polluted air masses. There is an average of ∼300 sunny days per year allowing at least 4 direct sun observations with the most
favorable clear sky conditions in the morning hours.

Davos is a small city (∼12'500 permanent inhabitants) in the adjacent valley south-east of Arosa at a mean altitude of 1550
m a.s.l. The industrial activity is more developed at Davos than at Arosa but is still limited. The measurement site (46.813 N,
9.844 E) is located in front of the PMOD/WRC building at 1590 m. PMOD/WRC sits above a well pronounced inversion layer
which effectively prevents local pollution from the valley to reach the Observatory.

Figure 2 illustrates the horizon as seen from the LKO site (blue shaded area) calculated from a model of the Swiss topog-
raphy at 25m horizontal resolution. The horizon seen from the PMOD/WRC site is illustrated by the red line. The blue lines
correspond to the course of the sun at the beginning of each month from January to June. The two horizons present similarities
with differences only at the extreme East and West for the Summer months where the Arosa site has a longer sun exposure.
But as the measurements with slant column (SC = ozone column * air mass = $O_3 \times \mu$) longer than ∼1000 are affected by the
stray light interference in the single monochromator instruments, the observations at Arosa are limited to air mass values $\mu \leq 4$
(*Christodoulakis et al.*, 2015). This corresponds to a SC of ∼1200 for the typical ∼300 DU ozone column of Arosa. At the
bottom of Figure 2, the six black lines correspond to the daily cycle of the $\mu$ values for the illustrated sun courses and the hori-
zontal line to the limit $\mu = 4$. The extra sun duration time for the Arosa site and $\mu$ values below this limit only occur during the
late afternoon hours in Summer. However, in the alpine environment clouds are frequently present at that time of the day and
the presence of trees at the North–West direction of LKO site prevent measuring until the sun reaches the horizon. Similarly to
Arosa, the number of days appropriate for direct sun measurements at Davos is ∼300 per year. Finally, the horizontal distance
between Arosa and Davos sites is ∼13 km and the altitude difference is ∼260 m.

In summary, the basic characteristics of the sites can be considered to be very similar which should, a priori, limit the
site-specific differences in the ozone column.

## 30   3   Instruments and analysis

For the comparison of the Arosa and Davos sites, the Mark II single monochromator Brewer $B_{072}$ from the Arosa triad was
moved to Davos and parallel measurements started on November $22^{nd}$, 2011. Besides a-six week interruption end of January
2013 due to technical problems, the operation has continued as intended.

Three "maintenance/calibration" campaigns with the European Brewer Calibration Center (RBCC-E) traveling reference
instrument have taken place during that period to assure the good working conditions of the different instruments . The first





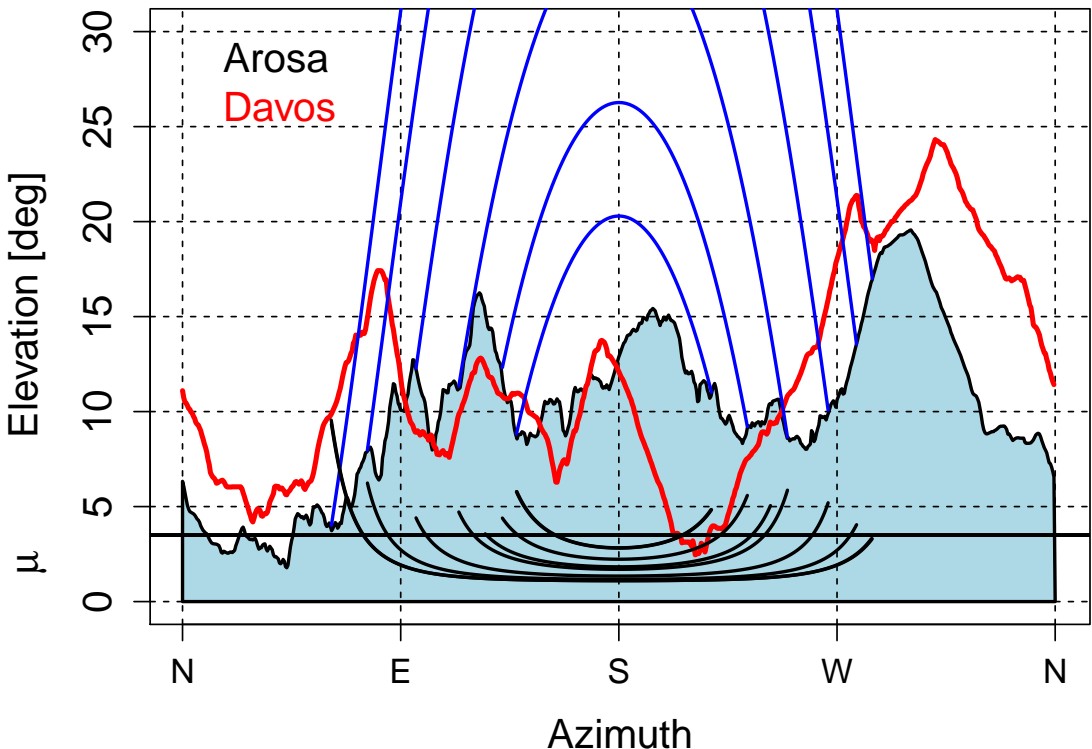

**Figure 2.** Calculated horizon from the Arosa (blue shaded) and Davos (red line) sites. The blue lines refer to the course of the sun at the beginning of the month from January to June. The black lines are the corresponding air mass $\mu$ daily cycle for the different sun elevation curves.

one happened in July 2012 and the second in July 2014 at LKO. Brewer $B_{072}$ was transported back to Arosa for this purpose for a short period in 2012 and a longer one in 2014 (see illustration in Figure 3). In the third campaign in 2016, the instrument

5    $B_{072}$ was calibrated in Davos and the instruments $B_{040}$ and $B_{156}$ subsequently in Arosa. To get a long term perspective of the changes of the instruments extra terrestrial constant (ETC) and the ozone absorption coefficient, we refer to Figure 2 of *Stübi et al.* (2017). In table 1, the recent changes of these parameters are reported. It can be seen that besides the change of ETC for Brewer $B_{072}$ in 2014, only minor corrections have been necessary to keep the instruments in agreement with the traveling reference instrument (*Redondas et al.*, 2015; *Redondas and Rodriquez-Franco*, 2015). The change of $B_{072}$ ETC resulted from

10    the improvement of the UV focusing, which significantly changed the response of the instrument (*Redondas and Rodriquez-Franco*, 2015).

    Details on the Arosa Brewer instruments are given in *Stübi et al.* (2017) therefore only a summary is presented here. The instruments were operated according to the standard acquisition and processing programs. The instrument constants were adjusted during the calibration campaigns and in between them, the data were reprocessed according to the time series of



**Table 1.** Parameters ETC and ozone absorption coefficient of the Arosa triad Brewer instruments from the last three calibration campaigns.

| Parameter | Validity period | $B_{040}$ | $B_{072}$ | $B_{156}$ |
|---|---|---|---|---|
| ETC | 2010–2012 | 2985 | 3168 | 1750 |
| | 2012–2014 | 2980 | 3188 | 1765 |
| | 2014–2016 | 2970 | 3230 | 1750 |
| | 2016– | 2970 | 3230 | 1750 |
| $O_3$ abs. coeff | 2010–2012 | 0.3335 | 0.3397 | 0.3326 |
| | 2012–2014 | 0.3335 | 0.3377 | 0.3402 |
| | 2014–2016 | 0.3335 | 0.3377 | 0.3390 |
| | 2016– | 0.3335 | 0.3377 | 0.3390 |

the internal lamps tests results. Only the direct sun observation were considered in this analysis and they were subject to an automatic and visual data screening on a daily basis, taking advantage of the presence of multiple collocated instruments at LKO, of ancillary surface radiation measurements and of cloud observations. Individual direct sun observations for each Brewer has to satisfy the standard deviation criteria ($\sigma_{O_3} \leq 2.5$ DU) and only data with air mass factor $\mu \leq 4.0$ were used in the present analysis.

For the comparison of Brewer total ozone data, coincident criteria were defined as time difference $\delta t \leq 5$ minutes and air mass difference $\delta \mu \leq 0.05$. Since the measurement program of all three Brewer instruments was driven by the same set of commands according to the solar zenith angle (SZA), 90% of the coincidences were within $\delta t \sim \pm 100$ seconds and $\delta \mu \sim \pm 0.016$.

Figure 3 presents the differences of the coincident measurements of the Brewer instruments $B_{040}$ and $B_{072}$ over the time period 2010–2016 in Dobson units (blue) and reported to the midday ozone column in [%] (red). This figure does not allow distinguishing any substantial break in the series resulting from the transfer of the Brewer $B_{072}$ instrument to Davos. During the time period 2010–2012 when the three instruments were collocated at Arosa, only a limited number of differences exceeded $\pm 5$ DU while this number increased when the instruments were apart. A slight seasonal cycle can also be noticed in this figure.

The parameters of the distribution of the differences series are given in table 2. The total number of coincident measurements were of the order of $\sim 50'000$, about half for collocation and half for distant locations as shown in the last column of table 2. The medians of the distributions showed a shift of the order of 0.25 % when Brewer $B_{072}$ was located at Davos accompanied by a widening of the inter-quantiles $Q_{75\%}$-$Q_{25\%}$ ($Q_{97.5\%}$-$Q_{2.5\%}$) of 0.2% (0.8%). The differences between two Mark II and between a Mark II and a Mark III was $\sim 0.2\%$, which could be a sign of stray light effect if these features present a seasonal component. The overall changes reported in table 2 correspond to differences of the order of one Dobson unit. To reveal such small differences, the instruments must to be very well maintained and calibrated, as provided during the campaigns.

In order to gain a clearer picture of the temporal evolution of the coincident data, the time series were aggregated to monthly median values. The resulting difference time series for the three pairs of Brewer instruments are shown in Figure 4 together




**Table 2.** Quantiles of the distributions of the differences expressed in [%] between pairs of Brewer instruments collocated at Arosa (period LKO) or with $B_{072}$ at PMOD/WRC (period PMOD).

| Brewer pair | Period | $Q_{2.5\%}$ | $Q_{25\%}$ | Median | $Q_{75\%}$ | $Q_{97.5\%}$ | $Q_{75\%}$-$Q_{25\%}$ | $Q_{97.5\%}$-$Q_{2.5\%}$ | Sample |
|---|---|---|---|---|---|---|---|---|---|
| $(B_{072}$-$B_{040})/B_{040}$ | LKO | -0.98 | -0.37 | -0.03 | 0.29 | 0.99 | 0.66 | 1.97 | 28191 |
| | PMOD | -1.07 | -0.19 | 0.23 | 0.68 | 1.68 | 0.87 | 2.75 | 25752 |
| $(B_{156}$-$B_{040})/B_{040}$ | LKO | -1.30 | -0.45 | -0.03 | 0.38 | 1.21 | 0.83 | 2.51 | 25643 |
| | PMOD | -1.18 | -0.33 | 0.12 | 0.56 | 1.47 | 0.89 | 2.65 | 25647 |
| | whole | -1.24 | -0.39 | 0.04 | 0.47 | 1.38 | 0.86 | 2.62 | 51290 |
| $(B_{072}$-$B_{156})/B_{156}$ | LKO | -1.40 | -0.50 | -0.09 | 0.34 | 1.28 | 0.84 | 2.68 | 20706 |
| | PMOD | -1.58 | -0.38 | 0.16 | 0.70 | 1.87 | 1.07 | 3.45 | 21153 |

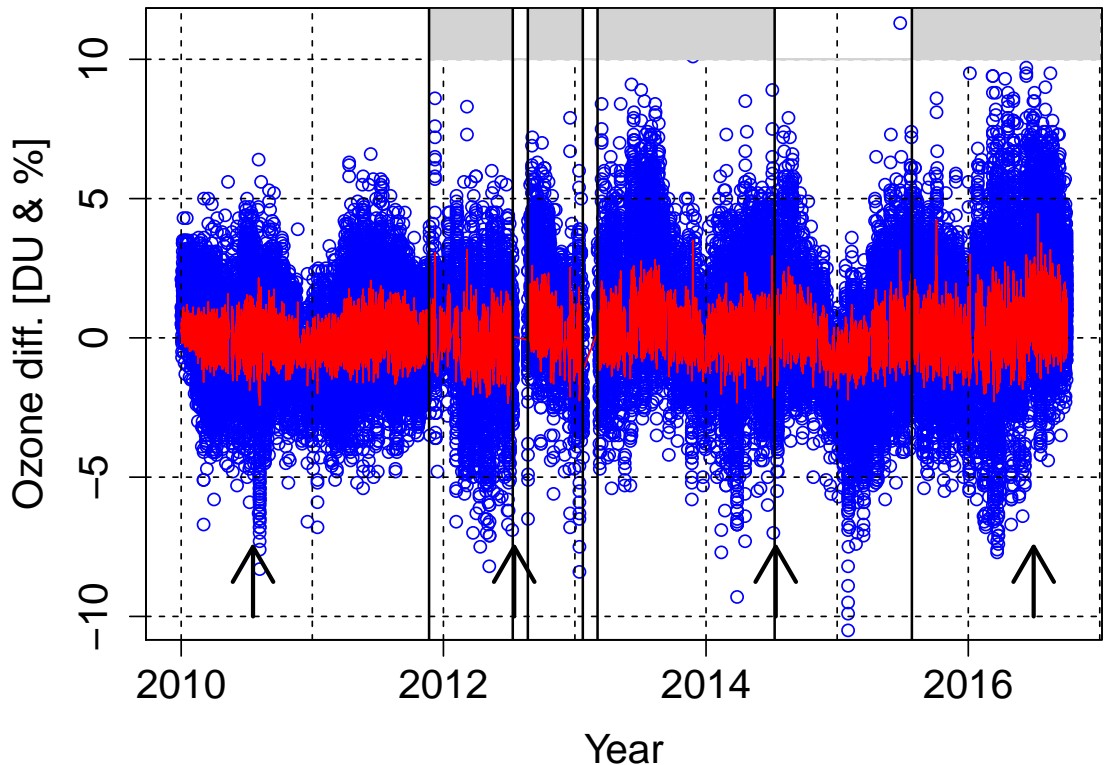

**Figure 3.** Time series of the differences between coincident measurements of the Brewer instruments $B_{040}$ and $B_{072}$ over the period 2010–2016. Blue: difference in DU units. Red: difference in [%]. The arrows indicate the calibration campaigns and the gray shaded areas denote the periods when Brewer $B_{072}$ were operated in Davos.





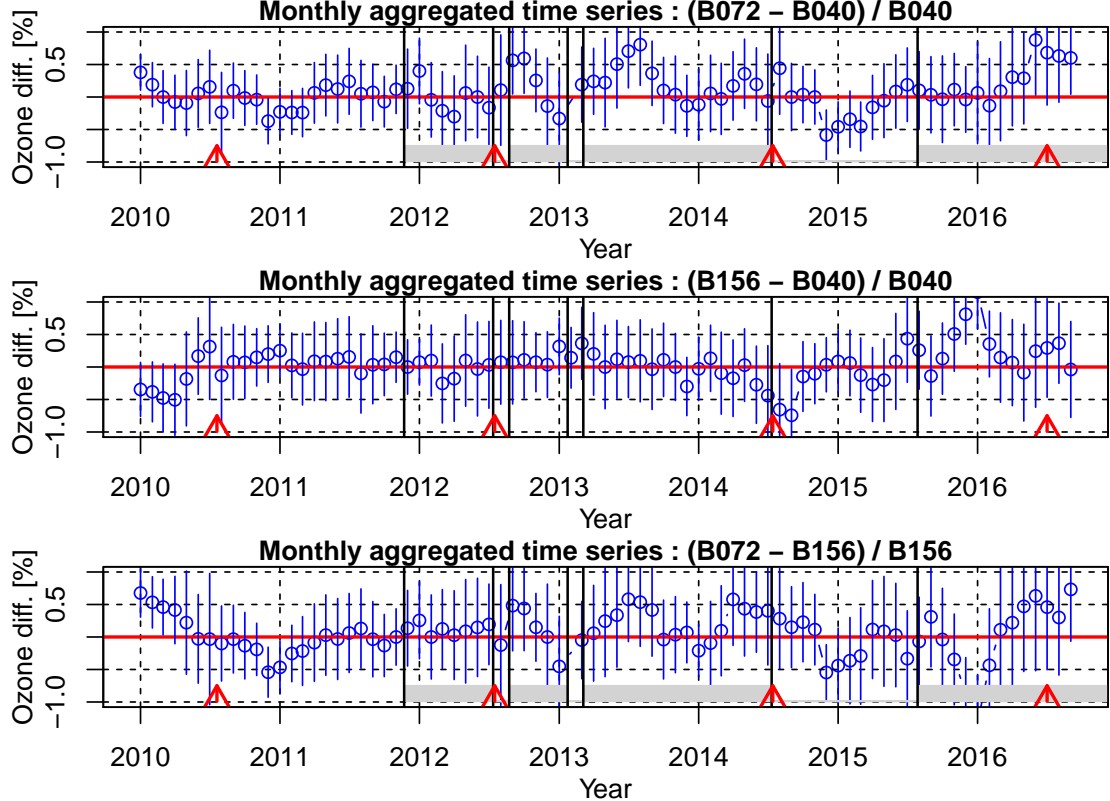

**Figure 4.** Time series of the monthly median differences of coincident ozone column measurements by pairs of Brewer instruments. The gray zones underline the periods when the Brewer $B_{072}$ instrument were located at Davos.

with the $2\sigma$ error bars. The monthly median differences between the collocated $B_{040}$ and $B_{156}$ instruments (middle panel) were mostly within the $\pm 0.5\%$ limits. On the lower and upper panels, the median variations were larger, touching or crossing the

5    $\pm 0.5\%$ limits more frequently and exhibiting larger variability.

To further characterize the data presented above, the time series were decomposed into a systematic and a random component on a daily basis following the method in *Stübi et al.* (2017). The first term is a measure of the long-term stability while the second term is linked to the short term variability. The pair of collocated Brewer instruments at LKO ($B_{040}$ and $B_{156}$) were considered as the reference. The diurnal variation of the ozone column was modeled as a fourth order polynomial function of

10   the time difference to 12 UTC fitted on all the measurements of the two reference Brewer instruments satisfying the quality and coincident criteria. A further constraint to avoid outliers and spurious coincidences was the requirement to have at least 5 coincident measurements for each Brewer instruments in the course of the day. Each Brewer was then characterized by an offset $\delta_{B_{nnn}}$ and a standard deviation $\sigma_{B_{nnn}}$ calculated as :

$$\delta_{B_{nnn}} = median\left[\sum_{i=1}^{n}[\{O_{3_{B_{nnn}}}\}_i - \{O_{3_{poly}}\}_i]\right] \tag{1}$$





$$\sigma_{B_{nnn}} = \sqrt{\sum_{i=1}^{n} \Big( [\{O_{3_{B_{nnn}}}\}_i - \{O_{3_{poly}}\}_i] - E\left[\{O_{3_{B_{nnn}}}\}_i - \{O_{3_{poly}}\}_i\right] \Big)^2 / (n-1)} \qquad (2)$$

$\{O_{3_{B_{nnn}}}\}_i$ being the measured data, respectively $\{O_{3_{poly}}\}_i$ being the model data and E[x] the mean value operator. As the ozone column varies smoothly on the time scale of a few hours, the value interpolated at noon was considered the ozone column representative for that day.

The offset $\delta_{B_{nnn}}$ represented the shift of the polynomial function to fit each instrument separately. The standard deviation $\sigma_{B_{nnn}}$ was a measure of the dispersion of the data around the smoothed daily variation represented by the polynomial function.

The fourth order polynomial as a model for the diurnal variation is not critical and alternative fit functions did not affect the results.

Figure 5 illustrates the procedure with the mean diurnal variation from the two reference instruments (black line) and the three individually fitted lines (broken lines) displaced for this particular day by the offsets: $\delta_{B_{040}}$ = -0.33 DU, $\delta_{B_{072}}$= 0.65 DU and $\delta_{B_{156}}$= 0.45 DU . The standard deviations for the three Brewer instruments amounted to $\sigma_{B_{040}}$= 1.2, $\sigma_{B_{072}}$= 1.1 and $\sigma_{B_{156}}$= 1.1 DU respectively.

The time series of the monthly median of $\delta$ and $\sigma$ terms divided by the ozone column at noon [%] for the three Brewer instruments are illustrated in Figure 6 for the period 2010–2016. The $\delta$ variations for $B_{040}$ and $B_{156}$ instruments mirror each other since they constituted the reference for the diurnal variation. They were below $\pm0.3\%$ for the entire period except at the end of 2015–beginning of 2016. The inter-quantiles ranges $Q_{97.5\%}$-$Q_{2.5\%}$ were only occasionally larger than 0.5% for these two instruments (error bars on Figure 6). The time series $\delta_{B_{072}}$ exhibited larger variations with extrema reaching $\pm0.7\%$ for the median. Similarly, the error bars for $B_{072}$ were larger as expected since this instruments is not part of the reference. These deviations of the $\delta_{B_{072}}$ series presented a structured pattern which required further attention.

For the $\sigma$ time series, the differences were much less pronounced reflecting the fact that $\sigma$ were an intrinsic factor of each instrument. The range of $\sigma$ for the $B_{040}$ and $B_{156}$ instruments was between $\sim0.2\%$ and $\sim0.5\%$ while for $B_{072}$, it was slightly larger by $\sim0.1\%$ but not significantly considering the $Q_{97.5\%}$-$Q_{2.5\%}$ inter-quantiles ranges. A distinct annual modulation in these $\sigma$ series were present for the three instruments.

In Table 3, the parameters of the distributions of the $\delta$ series are reported in the upper part and of the $\sigma$ series in the lower part. The $\delta$ series medians for the reference instruments were very close to zero while a shift of the $\delta_{B_{072}}$ distribution of $\sim0.2\%$ appeared when the instrument was located at Davos. The inter-quantile $Q_{97.5\%}$-$Q_{2.5\%}$ range of this last distribution was twice larger than the other single monochromator $B_{040}$ when $B_{072}$ instrument was collocated at Arosa and it widened to three times when it was located at Davos. While the two time periods should be indistinguishable for the $B_{040}$ and $B_{156}$ instruments, a variability of $\pm$ 0.1-0.2% of the different quantiles were observed.

The three $\sigma$ distribution parameters of the Brewer triad were very consistent with each other as seen in the lower part of table 3. The two reference Brewer instruments ($B_{040}$ and $B_{156}$) exhibited a similar median of $\sim0.35\%$. The inter-quantile





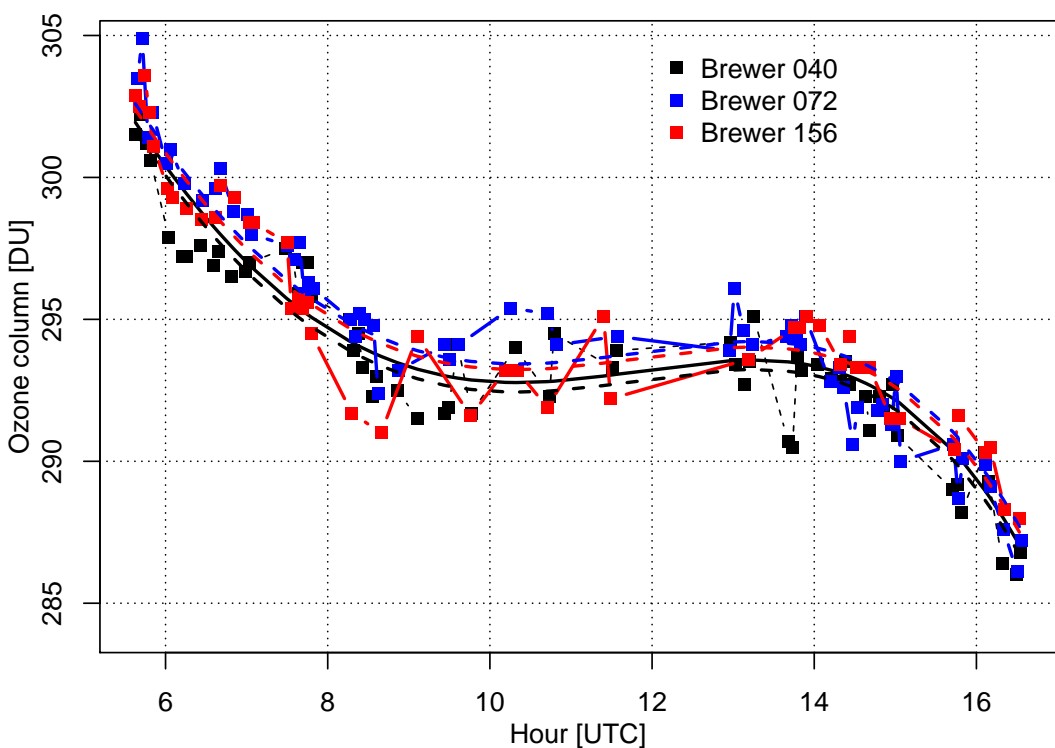

**Figure 5.** Diurnal variability of the Brewer triad for July 12, 2011: ozone column vs. time [UTC]. The continuous black line is the model diurnal variation and the dashed lines correspond to the model adjusted to each Brewer data set. The black, blue and red colors correspond respectively to $B_{040}$, $B_{072}$ and $B_{156}$ Brewer instruments.

range $Q_{97.5\%}$-$Q_{2.5\%}$ was ∼0.85% for the reference instruments. For Brewer $B_{072}$ instrument, the median was ∼0.40% and inter-quantile range $Q_{97.5\%}$-$Q_{2.5\%}$ were, respectively 0.96% and 1.06% for the LKO and the PMOD/WRC periods.

5    Stray light effect are known to affect the measurements at long path which depend on the season and the amount of ozone. This effect concerns essentially single monochromator instruments (Mark II: $B_{040}$ and $B_{072}$) and it varies from instrument to instrument. This phenomenon was tested by considering three groups of $\delta$ values according to the SC value as illustrated in Figure 7 which shows the seasonality of the SC distribution averaged over the period 2011-2017. From April to October the SC was below 800 for 90% of the data which were essentially free of the stray light effect as is the case for most of the Brewer

10   instruments. Contrarily, measurements in December and January showed SC values above 800 which may potentially have induced a low ozone column bias for the two single monochromator instruments $B_{040}$ and $B_{072}$. To evaluate the importance of this effect, the $\delta$ time series was grouped in three classes of SC according to the median of the monthly SC distributions illustrated in Figure 8 by the red lines. In the upper panel corresponding to the Arosa period with three collocated instruments, only Brewer $B_{072}$ showed a significant SC dependence. Brewer $B_{040}$ had a better stray light rejection and presented no SC





**Table 3.** Quantiles of the distribution of the parameters $\delta$ and $\sigma$ of the three Brewer instruments express in [%]. Under the "Period" column, LKO means that the $B_{072}$ instrument was collocated at Arosa, respectively PMOD/WRC correspond to having $B_{072}$ at Davos and "All" for the whole period. The sample sizes were 705 days for the Arosa period and 801 days for the Davos period.

| Parameter | Period | $Q_{2.5\%}$ | $Q_{25\%}$ | Median | $Q_{75\%}$ | $Q_{97.5\%}$ | $Q_{75\%}$-$Q_{25\%}$ | $Q_{97.5\%}$-$Q_{2.5\%}$ |
|---|---|---|---|---|---|---|---|---|
| $\delta B_{040}$ | LKO | -0.31 | -0.09 | 0.01 | 0.10 | 0.35 | 0.19 | 0.66 |
| | PMOD | -0.50 | -0.16 | -0.04 | 0.06 | 0.26 | 0.22 | 0.76 |
| $\delta B_{156}$ | LKO | -0.41 | -0.13 | -0.02 | 0.11 | 0.32 | 0.26 | 0.73 |
| | PMOD | -0.33 | -0.08 | 0.04 | 0.18 | 0.55 | 0.26 | 0.88 |
| $\delta B_{072}$ | LKO | -0.70 | -0.20 | 0.03 | 0.21 | 0.66 | 0.41 | 1.36 |
| | PMOD | -0.77 | -0.09 | 0.21 | 0.52 | 1.07 | 0.61 | 1.84 |
| $\sigma B_{040}$ | All | 0.17 | 0.28 | 0.34 | 0.41 | 0.57 | 0.69 | 0.84 |
| $\sigma B_{156}$ | All | 0.18 | 0.30 | 0.37 | 0.45 | 0.61 | 0.75 | 0.89 |
| $\sigma B_{072}$ | LKO | 0.19 | 0.31 | 0.39 | 0.48 | 0.77 | 0.79 | 0.96 |
| | PMOD | 0.21 | 0.35 | 0.44 | 0.55 | 0.85 | 0.90 | 1.06 |

dependence. The lower panel corresponding to the Davos period, the three $B_{072}$ box-plots showed the similar SC induced pattern moreover slightly shifted upwards.

5    The seasonal variation can be better analyzed on aggregated $\delta$ time series in monthly medians as illustrated in Figure 9. The upper panel is for the months with the three Brewer instruments collocated at Arosa and the lower panel for the Davos period. The errors bars are the inter-quantiles ranges $Q_{97.5\%}$-$Q_{2.5\%}$. No seasonal cycle can be distinguished on the $\delta_{B_{040}}$ and $\delta_{B_{156}}$ monthly median series neither on the upper panel nor on the lower panel. The Brewer $B_{072}$ aggregated $\delta_{B_{072}}$ series present a discernible seasonal component at both location even if it is not significant at the 95% level.

10   The altitude difference between Arosa and Davos of 260 m introduces a difference of total column ozone above the two sites. Unfortunately, without actual measurements, it is impossible to characterize the ozone column above Davos from the surface to 260 m above ground, but surface ozone observations at the two locations and a comparison with free tropospheric observations can be used to derive meaningful boundaries on the magnitude of this difference. At Arosa, surface ozone observations are conducted by the local authorities at the LKO site. At Davos, a surface ozone station is operated as part of the Federal air

15   pollution monitoring network (NABEL). This station is nine hundred meters away from the PMOD/WRC site on a 35 m high tower in a forest. As discussed by *Chevalier et al.* (2007) for the central European region, ozone concentrations in the free troposphere increase with altitude fairly consistently irrespective of geographical location, however, this stratification can be modulated substantially by surface effects. Thus, surface ozone observations made at stations below 1200 m a.s.l were found to be as much as 40% lower than those observed by aircraft and ozone balloon sondes in the free troposphere. The deviations decreased with increasing altitude of the stations to less than 8% for stations above 2000 m a.s.l. The mean annual surface ozone





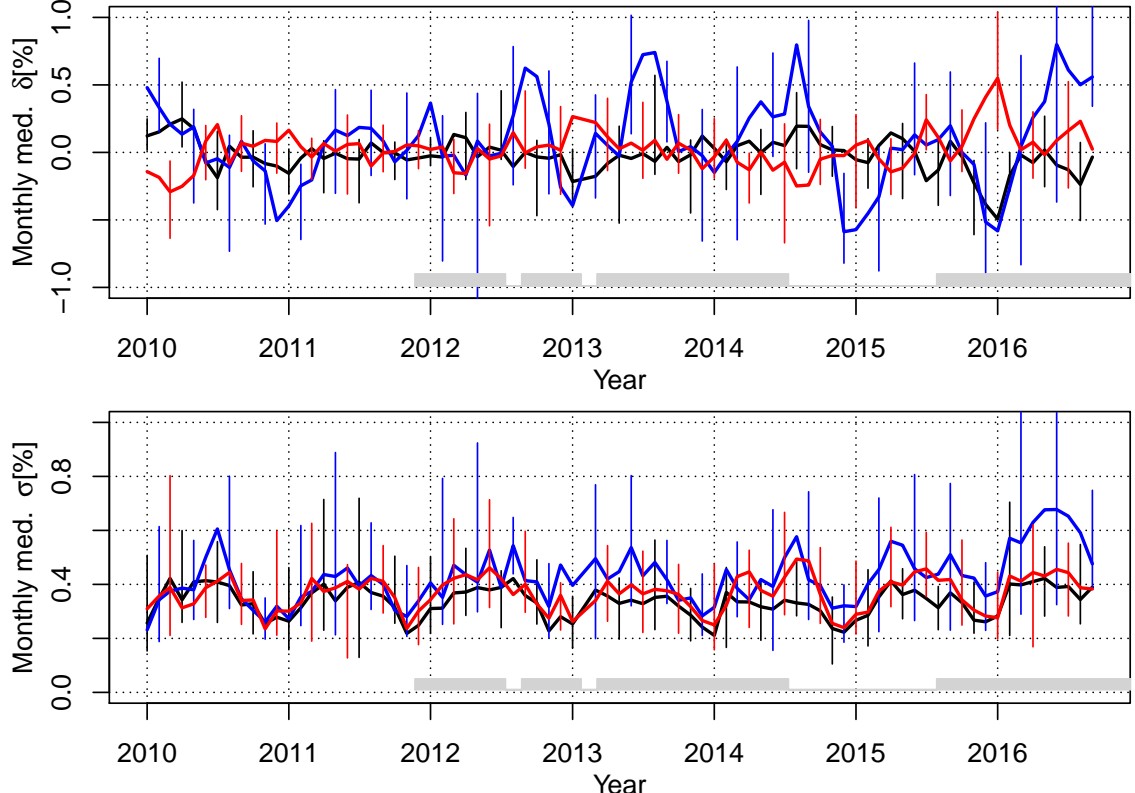

**Figure 6.** Time series of the monthly median offset $\delta$ (upper panel) and median standard deviations $\sigma$ (lower panel) over the time period 2010–2016. The black, blue and red color lines correspond respectively to Brewer $B_{040}$, $B_{072}$ and $B_{156}$ instruments. The error bars corresponding to the inter-quantiles $Q_{97.5\%}$-$Q_{2.5\%}$ ranges are displayed every third month for clarity. The gray areas show the time periods when the Brewer $B_{072}$ instrument was located at Davos.

concentrations observed at Arosa (42.3±8.2 ppb) and Davos (42.0±7.1) were found to be very similar despite the difference in altitude (see Figure 4 in *Chevalier et al.* (2007)). Both stations exhibit about 5 ppb lower ozone concentrations than expected from free tropospheric profiles, with hourly variabilities of ∼11 ppb in Summer and ∼8 ppb in Winter. The seasonal variability was on the order of 15-20 ppb. This variability is likely due to changing weather and synoptic scale transport, with super-imposed photochemical effects of local pollution.

Under the hypothesis of a constant ozone mixing ratio above the inversion layer mentioned in section 2, the partial ozone column can be estimated as:

$$\Delta\,Column\,O_3(t) = \int\limits_{p_{Davos}(t)}^{p_{Arosa}(t)} O_3(t,p)*dlnp = 0.79*\{\ln(p_{Davos}(t)) - \ln(p_{Arosa}(t))\}*O_3(t) \qquad [DU] \qquad (3)$$





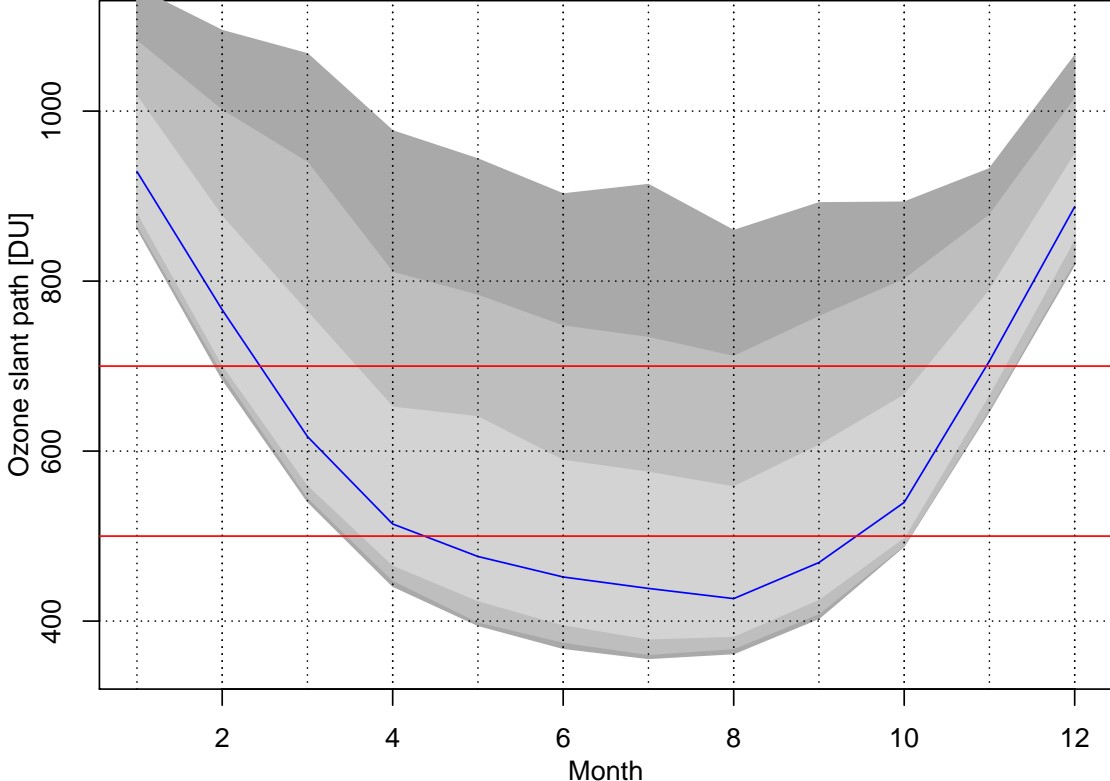

**Figure 7.** Seasonal cycle of the slant path over Arosa. The different bands from light gray to dark gray encompass the respective SC limits of $Q_{97.5\%}$ - $Q_{2.5\%}$, $Q_{90\%}$ - $Q_{10\%}$ and $Q_{75\%}$ - $Q_{25\%}$. The blue line is for the monthly medians.

with the ozone $O_3(t)$ [ppm] values measured at Davos NABEL surface station. The result of this simple calculation based on the hourly daytime measurements over the time period 2006-2014 is shown in Figure 10 and suggests an average contribution

5    of the partial ozone column to the total ozone column above Davos of $\sim$0.25%$\pm$0.15%.

## 4   Discussion

The Brewer sun spectrophotometers are very reliable instruments to measure the ozone column in a fully automated mode within 1% reproducibility. It is therefore well adapted to compare two different sites under the same measurement program. This allowed us to generate a large dataset of coincident measurements to evaluate the potential impact of the instrument

10   location on the measurements during an extended period of time. This approach is in line with the Global Climate Observing System (GCOS) monitoring principles 1 and 2 (*GCOS* (2003)) which state that an assessment of impact should be made before the implementation of a change and a suitable period of overlap of the observing systems is required. It has been applied here to document a possible move of the Arosa based ozone column monitoring activities at the nearby Davos site.



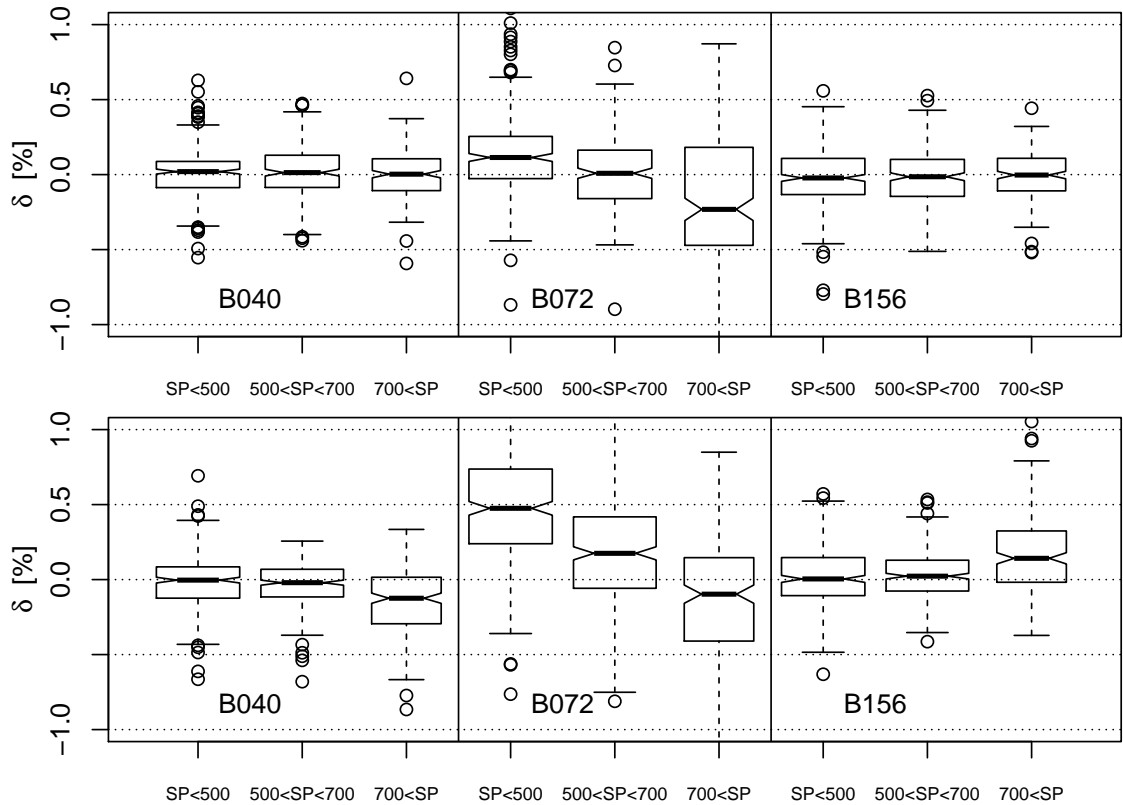

**Figure 8.** Boxplot of the $\delta$ time series of the three Brewer instruments according to three classes for the median of the ozone slant path monthly distributions (red lines in Figure 7): median SC $\leq$ 500, 500 < median SC $\leq$ 700 and 700 < median SC. The upper panel is for the Arosa period, respectively the lower panel for the Davos period and from left to right, the distributions of respectively $\delta_{B_{040}}$, $\delta_{B_{072}}$ and $\delta_{B_{156}}$.

The alignment of the Arosa Brewer instruments with respect to the global network is assured by the regular maintenance and calibration campaigns which have demonstrated an agreement with the RBCC-E traveling reference to within $\pm1\%$ and

5   even closer to within $\pm0.5\%$ in the recent years. The present analysis was concentrated on the study of the relative difference between the measurements obtained from the traveling Brewer instrument $B_{072}$ located either at Arosa or at Davos and the two other instruments of the Arosa Brewer triad ($B_{040}$ & $B_{156}$).

The large number of direct coincident measurements reported in table 2 and illustrated in Figure 3 showed that no significant differences were present at this stage. It was not possible to identify minor influences of the sites and a further step in the

10  analysis was required. It consisted in a separation in two terms: the short term variability $\sigma$ (random term) and the longer term stability $\delta$ (systematic term). The same approach was already used in *Stübi et al.* (2017) to evaluate the quality of the Arosa Brewer triad measurements. The findings were a stability of the instruments of $\sim 0.40\%$ over a time scale of a decade and

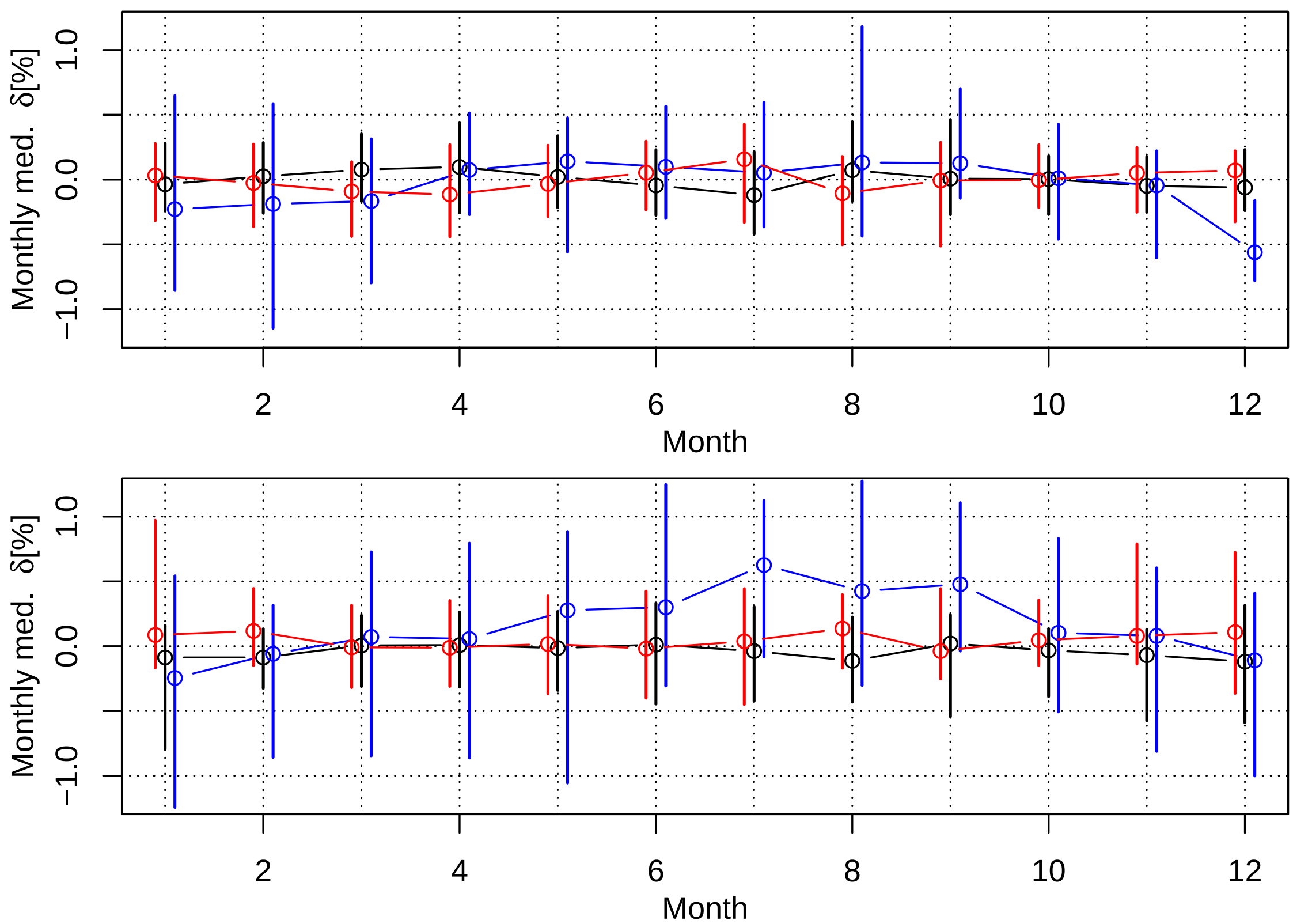

**Figure 9.** Seasonal cycle of the offset $\delta$ for respectively Brewer $B_{040}$, $B_{072}$ and $B_{156}$ instruments in red, blue and green color, respectively. The upper panel corresponds to the Arosa period while the lower panel corresponds to the Davos period.

a short term variability of ∼0.3%, in good agreement with a similar analysis of the world reference Brewer triad at Toronto (*Fioletov et al.*, 2005; *Kerr and al.*, 1998).

The results of this method are presented in table 3 and Figure 6. The three $\sigma$ distributions present a seasonal variation of ∼0.2%. The larger air mass factors prevailing in Winter induced a larger difference of the measurement signals and the extra terrestrial constant and potentially a better signal/noise ratio thus, reducing $\sigma$. Contrarily, the frequent transitions between imperfect neutral density filters in Summer may be a source of additional fluctuations impacting the short term variability. The distribution of $\sigma_{B_{072}}$ is slightly larger than the other two, independently of the sites.

Regarding the $\delta$ parameter for the Arosa period, a larger $\delta_{B_{072}}$ distribution was expected since it was not part of the reference built upon $B_{040}$ and $B_{156}$ instruments data. A factor $\sqrt{3}$ could be expected based on assumption that instrumental uncertainty are the same for all instruments and not correlated. For the Davos period, a shift (∼0.2%) and a widening (factor ∼1.4) of the $\delta_{B_{072}}$ distribution was observed.

Two additional factors were investigated that could potentially make a contribution to differences between measurements at the two investigated sites. First, the altitude difference created an additional partial column which is estimated of the order





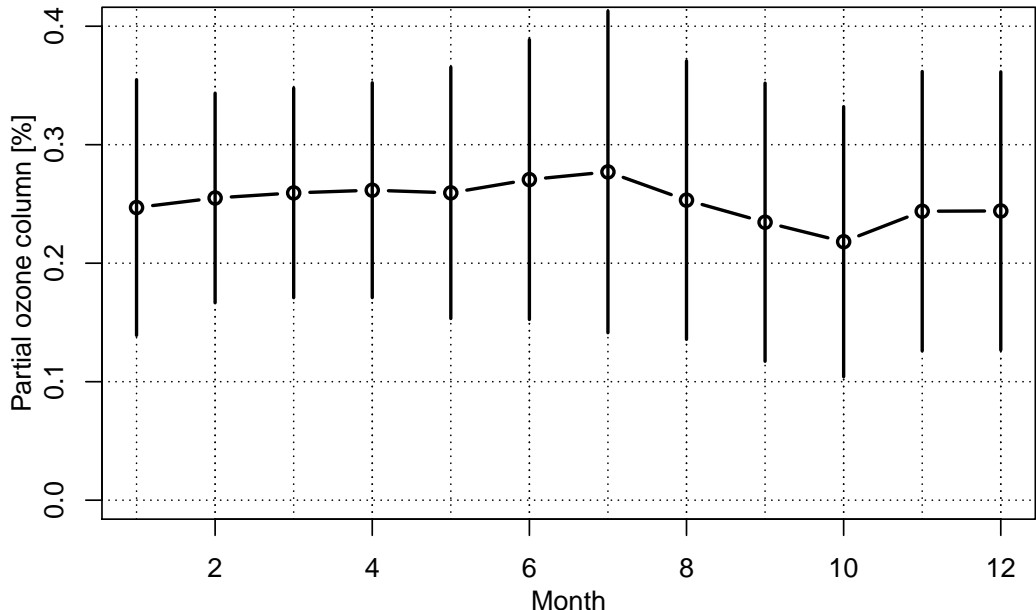

**Figure 10.** Monthly mean percentage contribution of the partial ozone column integrated from the Davos to Arosa based on the NABEL surface ozone measurements to the total column. The bars correspond to two standard deviations.

of ∼0.25% (∼0.8 DU) based on simple assumptions. This partial column is of the same order of magnitude as the median of the $\delta_{B_{072}}$ distribution for the Davos time period of $B_{072}$ instrument. Secondly, the upper panel of Figure 9 shows a seasonal

5    curvature in the $\delta_{B_{072}}$ distribution which had the signature of a stray light problem. The investigation confirmed (see Figure 8) a pronounced bias for Brewer $B_{072}$ for the Winter months which could explain the mentioned curvature.

In summary, the present analysis did not show significant differences between coincident Brewer measurements at the two locations. The locations Davos and Arosa did not introduce discontinuities in the measurements of the ozone column. A small seasonal variation of the ozone column difference of ± ∼0.2% was identified in the analysis which could not be distinguished

10   from a stray light bias of the test $B_{072}$ instrument. The altitude difference between the two sites generated a non-significant partial column based on surface ozone measurements embedded in the total uncertainties of the measurements .

Finally, the analysis of coincident data precluded the possibility of observing differences in the sampling of the daily cycle at the two sites, e.g. larger numbers of observations at low sun angles. The analysis of the horizons from Arosa and Davos sites was checked with the help of a high resolution model of the topography. A bias of the ozone column due to a different sampling of the diurnal cycle caused by the topography of the sites was excluded.



## 5 Conclusions

The motivation for this work was the perspective of a continuation of the Arosa total ozone column series at the Davos site. The

analysis of the campaign data between the two stations separated horizontally by 13 km and vertically by 260 m showed that the two sites agree to better than ∼0.5%, which represents the limit of the combined long term stability and the precision of Brewer instruments. The monthly median differences are of the order of ∼0.2%, superimposed with an additional seasonal cycle of the same order, none of these numbers being significant at the 95% confidence level. On the annual mean, the measurements from the two sites agreed within less than 0.25%. It remains speculative if the altitude difference could be the reason of the small

but not significant difference on the ozone column. The quality of the Arosa data set and especially its continuity has always been a great concern in the maintenance and development of the LKO observatory. From this analysis, it can be concluded that the ozone column series initiated at Arosa in 1926 would not been disrupted by a change of site. Local factors potentially influencing the measurements are below the measurements variability and stay below the long term stability of the Brewer instruments and within the uncertainties associated to the calibration procedures of the Brewer network. The confirmation of

these results with automated Dobson instruments is presently underway and the results will be reported after completion of this second parallel measurements campaign

## 6 Data availability

The data used for this analysis are available at the WOUDC for the Brewer $B_{040}$ instrument. The complete data sets can be requested by direct contact with the corresponding author. In a near future, they all will be available at the EUBREWNET data

center (see European Cooperation in Science and Technology: COST ES 1207: A European Brewer Network (EUBREWNET), http://www.eubrewnet.org).

*Author contributions.* R. Stübi has made the analysis of the data and written the first version of the manuscript. H. Schill was in charge of the quality control and the preparation of the data sets. L. Egli was responsible for the daily control of the Brewer $B_{072}$ in Davos. J. Klausen, L. Vuilleumier, J. Gröbner, L. Egli and D. Ruffieux have revised the manuscript.



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
