# Peer review of "On the compatibility of Brewer total column ozone measurements in two adjacent valleys (Arosa and Davos) in the Swiss Alps"

_Atmospheric Measurement Techniques, 2017_

## Referee Comment (RC1) · Anonymous Referee #2 · 17 Jul 2017

The manuscript presents a very important study which is necessary to justify the relocation of the historic total ozone station of Arosa to Davos. The authors have made a very detailed analysis which fully supports the derived conclusion. Obviously the comparison of the data at the two locations is very good and any differences observed are minor and arise mostly from the instruments' uncertainty. The paper is well presented; hence I recommend to be accepted for publication in AMT. The comments below are mainly for improving the quality and readability of the paper.

General comments:

The discussion section could well stand for "conclusions" as it summarizes the findings

of the study. Therefore, the conclusions section is a more brief repetition of the discussions section. I suggest to consider the possibility of merging the two sections as "Discussion and conclusions".

Some of the figures show markers with error bars but not in all of them is defined whether they correspond to one or two standard deviations. This becomes confusing when in Fig. 10 the error bars represent 2 standard deviations. Please make figures consistent.

Specific comments:

4, 19-21: This sentence is not well written and could confuse the inexperienced reader. Also confusing is the use of two SC values (1000 and 1200). I am not sure if the introduction of the "slant column" makes any sense here. I would suggest to move it later to page 10 where SC is actually used and rephrase the sentence to something like:

"However, the longer sun exposure of Arosa is not important as the observations are limited to air mass values $\mu \leq 4$ (Christodoulakis et al., 2015) in order to reduce the effect of the stray light interference in the single monochromator Brewer instruments. "

4, 32: Please clarify "a-six week interruption end of January 2013": Is it "a six-week interruption starting at the end of January 2013" or something different?

5, 8: Please specify briefly what other (except the ETC ) minor corrections have been necessary.

11, 3-4: I miss some discussion on the effect of stray light. This section discusses the problem and presents a method to determine the stray light effect, but it ends without addressing the importance of this effect on the comparison results.

11, 9: Can you elaborate on the causes of observed seasonal component in the differences and standard deviation of B072?

15, 7-8: This is the first time that the effect of neutral density filters is mentioned. I would suggest to expand briefly the discussion so that an inexperienced reader can understand the topic.

Technical comments:

1, 17: Replace "From the 1920s onwards" to "Since the 1920s"

1, 21: Replace "low polluted" to "low-pollution"

2, 9: Replace "have" to "has" (assuming that "series" refers to one series)

2, 12: Replace "The ozone hole problem" with "The depletion of stratospheric ozone"

2, 13: Replace "of the total ozone" with "in the total ozone"

2, 14: Replace "consequences" with "effects", as consequences implies a negative effect which is not true for the MP.

3, 2: Replace "before the 1970s, respectively after the 1997s." with "before 1970 and after 1997, respectively."

3, 14: Replace "section 4 and followed" with "section 4, followed"

4, 11: Replace "12'500" with "12,500"

4, 19-21: Slant columns ($\sim$1000 and $\sim$1200) are not unitless; please use DU.

5, 3: Remove "one"

5,6: Replace "instruments" with "instruments'"

5,7: Insert "the" before "ETC"

6,1: Delete "Parameters"

7, last line: Replace "were operated" with "was operating"

9, 19: Replace "extrema" with extremes"

9, 20: Replace "instruments" with "instrument"

10, 5: Replace "are" with "is"

10, 13: There are no "red lines" in Figure 8.

11, 9: Replace "location" with "locations"

---

## Author Comment (AC1) · 26 Sep 2017

**Reply to reviewer # 1 comments on the manuscript AMT-2017-158**

The reviewer considered our manuscript to be accepted for publication with minor revisions. We thank the reviewer for the careful reading and the suggestions that allows us to improve our manuscript.

**Referee RC 1**

**Comment 1**  The discussion section could well stand for "conclusions" as it summarizes the findings of the study. Therefore, the conclusions section is a more brief repetition of the discussion section. I suggest to consider the possibility of merging the two sections as "Discussion and conclusions".

**Reply 1**  We agree with this comment that there are many repetitions in the two sections. However from our own experience, many readers like to read the more synthetic conclusions with lower interest to details given in the discussions. Therefore, we prefer to keep the two sections separated.

**Comment 2**  Some of the figures show markers with error bars but not in all of them is defined whether they correspond to one or two standard deviations. This becomes confusing when in Fig. 10 the error bars represent 2 standard deviations. Please make figures consistent.

**Reply 2**  The definition of the errors bars has been added when missing e.g. in Figures 4 and 9. We have used the standard deviations where the distributions were close to normal and the interquartile where the distribution were skewed.

**Comment 3**  4, 19-21: This sentence is not well written and could confuse the inexperienced reader. Also confusing is the use of two SC values (1000 and 1200). I am not sure if the introduction of the "slant column" makes any sense here. I would suggest to move it later to page 10 where SC is actually used and rephrase the sentence to something like: "However, the longer sun exposure of Arosa is not important as the observations are limited to air mass values __4 (Christodoulakis et al., 2015) in order to reduce the effect of the stray light interference in the single monochromator Brewer instruments. ".

**Reply 3**  The suggested sentence from the referee is clearer and we agree that the introduction of the slant column concept is more appropriate at page 10. We have introduced the suggested sentence on page 4 and extended the stray light discussion on page 10.

**Comment 4**  4, 32: Please clarify "a-six week interruption end of January 2013": Is it "a six-week interruption starting at the end of January 2013" or something different?

**Reply 4**  We have corrected the sentence introducing the exact time period which were in fact 5 weeks long.

**Comment 5**  5, 8: Please specify briefly what other (except the ETC ) minor corrections have been necessary.

**Reply 5**  Changes of the dead time and the temperature coefficients have been added at page 5. These are the other two main parameters influencing the ozone measurements.

**Comment 6**  11, 3-4: I miss some discussion on the effect of stray light. This section discusses the problem and presents a method to determine the stray light effect, but it ends without addressing the importance of this effect on the comparison results.

**Reply 6**  Additional sentences have been added to develop this point at pages 10-11 (see also comment 7 of the second referee)

| | |
|---|---|
| Comment 7 | 11, 9: Can you elaborate on the causes of observed seasonal component in the differences and standard deviation of B072? |
| Reply 7 | We suggested in the previous paragraph that the stray light induced low ozone bias in winter produces an annual cycle in the difference. The discussion on the cause of the annual variation on the standard deviations is also mentioned in page 15-16. |
| Comment 8 | 15, 7-8: This is the first time that the effect of neutral density filters is mentioned. I would suggest to expand briefly the discussion so that an inexperienced reader can understand the topic. |
| Reply 8 | We have added a sentence explaining the use of the neutral density filters in the Brewer which could be a potential source of the increase of the standard deviation in summer (comment 7 above). |
| Comment 9 | Technical comments |
| Reply 9 | All technical comments have been introduced in the manuscript. We thank the referee for the careful reading of the manuscript. |
| Comment 10 | 1, 17: Corrected "From the 1920s onwards" to "Since the 1920s" |
| Comment 11 | 1, 21: Corrected "low polluted" to "low-pollution" |
| Comment 12 | 2, 9: Corrected "have" to "has" (assuming that "series" refers to one series) |
| Comment 13 | 2, 12: Corrected "The ozone hole problem" with "The depletion of stratospheric ozone" |
| Comment 14 | 2, 13: Corrected "of the total ozone" with "in the total ozone" |
| Comment 15 | 2, 14: Corrected "consequences" with "effects", as consequences implies a negative effect which is not true for the MP. |
| Comment 16 | 3, 2: Corrected "before the 1970s, respectively after the 1997s." with "before 1970 and after 1997, respectively." |
| Comment 17 | 3, 14: Corrected "section 4 and followed" with "section 4, followed" |
| Comment 18 | 4, 11: Corrected "12'500" with "12,500" |
| Comment 19 | 4, 19-21: Slant columns (_1000 and _1200) are not unitless; please use DU. This has been introduced. |
| Comment 20 | 5, 3: Remove "one" |
| Comment 21 | 5,6: Corrected "instruments" with "instruments"' |
| Comment 22 | 5,7: Insert "the" before "ETC" |
| Comment 23 | 6,1: Delete "Parameters" |
| Comment 24 | 7, last line: Corrected "were operated" with "was operating" |
| Comment 25 | 9, 19: Corrected "extrema" with extremes |
| Comment 26 | 9, 20: Corrected "instruments" with "instrument" |
| Comment 27 | 10, 5: Corrected "are" with "is" |
| Comment 28 | 10, 13: There are no "red lines" in Figure 8. These lines have been added. |
| Comment 29 | 11, 9: Corrected "location" with "locations" |

---

## Author Comment (AC2) · 26 Sep 2017

**Reply to referee #2 comments on the manuscript AMT-2017-158**

The reviewer considered our manuscript to be accepted for publication with minor revisions. We thank the reviewer for the careful reading and the suggestions that allows us to improve our manuscript.

**Referee RC 2**

**Comment 1**   There is however, one issue, which is not addressed and should be discussed as an amendment: The long term Arosa TOC record is based on Dobson measurements since 1926, the investigation of the compatibility is done with a Brewer, which are in operation since 1998. A brief section describing the principal differences and/or good agreement between Dobsons and Brewers would help to accept, that the presented good results for Brewer observations can be transferred to the Dobson long term record too.

**Reply 1**   We have addressed this point in the discussion by pointing to the studies by Scarnato et al. 2014 and Redondas et al. 2014 who have both studied the comparison of the Brewer and Dobson data from Arosa. The reference to the presently running campaign of comparison of the two sites with Dobson instruments was already mentioned in the conclusion.

**Comment 2**   1,19: An additional reference of Dobson, 1968 (Forty Years' Research on Atmospheric Ozone at Oxford: a History. March 1968 / Vol. 7, No. 3 / APPLIED OPTICS) will complete the references.

**Reply 2**   The reference has been added and it appears now in line 19, p. 1.

**Comment 3**   2,2: Here an additional reference of a relevant Dütsch publication is recommended.

**Reply 3**   A reference to the 1984 article from Dütsch has been added at page 2, line 5.

**Comment 4**   2,6: The citation of Scarnato 2009 does not make sense at this point.

**Reply 4**   The citation has been removed.

**Comment 5**   2,17: The reference Yang et al. refers to Antarctic ozone recovery, but not to the LKO series.

**Reply 5**   The reference was cited as an example of analysis showing the different stages of the ozone layer recovery but not as a reference to the Arosa series. We agree that it was confusing and so we have removed it.

**Comment 6**   2, 19 + 18,18: Kerr an Mc Elroy is indeed published in 1995 and not in 1989

**Reply 6**   The reference has been corrected.

**Comment 7**   As reason for the limitation of observations (mue-values less than 4) stray light interference in the single monochromator is mentioned. This is an incomplete explanation: two straylight effects have influence on the TOC observations. Internal straylight in the instrument (especially in single monochromators like Brewer Mk II) is caused by misrouted light mostly of longer wavelengths. The discussed muedepending limitation effect, however, is mainly caused by external stray light. This means, that light comes from the sky around the sun disc, which has a different spectral composition (larger fraction of longer wavelengths) than direct sunlight. The hazier sky is the larger is this effect, resulting in a drop of the TOC value at low sun. Dobsons with a larger field of view around the sun (8 degree) than the Brewer (3 degree) are stronger affected and show an earlier drop of the TOC than Brewers (even than the single monochromators). This relation should be described a little more in detail. Whether the lower altitude in Davos (1590 m) than in Arosa (1850 m) with potentially larger turbidity leads to a lower mue-limit for good measurements cannot be stated, but is rather supposable in the Dobson data than in Brewer observations.

Reply 7    A new paragraph has been added to better introduce the stray light problem.

Comment 8   The citation of at least one reference for the different straylight effects would be helpful.

Reply 8    References to the stray light discussion have been added too.